# Developing an Integrated Framework for Securing Internet of Things Traffic in Smart Cities Using Machine Learning Techniques

**Moody Alhanaya and Khalil Al-Shqeerat ***

Department of Computer Science, College of Computer, Qassim University, Buraydah 51452, Saudi Arabia;
m.alhanaya@qu.edu.sa
* Correspondence: kh.alshqeerat@qu.edu.sa

**Abstract:** Internet of Things technology opens the horizon to a broader scope of intelligent applications in smart cities. However, the massive amount of traffic exchanged among devices may cause security risks, especially when devices are compromised or vulnerable to cyberattack. An intrusion detection system is the most powerful tool to detect unauthorized attempts to access smart systems. It identifies malicious and benign traffic by analyzing network traffic. In most cases, only a fraction of network traffic can be considered malicious. As a result, it is difficult for an intrusion detection system to detect attacks at high detection rates while maintaining a low false alarm rate. This work proposes an integrated framework to detect suspicious traffic to address secure data communication in smart cities. This paper presents an approach to developing an intrusion detection system to detect various attack types. It can be carried out by implementing a Principal Component Analysis method that eliminates redundancy and reduces system dimensionality. Furthermore, the proposed model shows how to improve intrusion detection system performance by implementing an ensemble model.

**Keywords:** Internet of Things; machine learning; intrusion detection system; ensemble classifier; Principal Component Analysis

## 1. Introduction

Internet of Things (IoT) technology significantly increases the efficiency and productivity of smart cities. However, they must be protected against potential security threats and attacks. Whenever a cyberattack escalates, security becomes a major concern. In order to protect insightful information, security systems identify anomalies. Unfortunately, security measures like encryption techniques and firewalls have been deployed to protect the system, but several attacks have bypassed them. Therefore, it is essential to recognize these attack patterns earliest to avoid losses to critical resources. Accordingly, appropriate actions can be taken to remove this intrusion.

A computer intrusion detection system (IDS) is one of the most effective tools for detecting attempts to access, manipulate, or shut down a computer system in an undesirable manner. It monitors the entire network's traffic from input and output, including resource activity, to protect the system from attacks [1]. IDS is an essential tool for designers of secure network systems, without which a considerable amount of data cannot be scanned in a second. One of the most promising approaches to improve the performance of IDSs is using machine learning (ML). This method can be used for both anomaly detection and misuse detection. In addition, an IDS can identify malicious and benign traffic by analyzing network traffic.

In most cases, only a fraction of the traffic that passes through a network can be considered malicious. This situation makes it hard for an IDS to identify attacks with high attack detection rates while keeping the false alarm rate low. One of the main challenges that IDSs face is the lack of ML models used to build an effective IDS. Researchers have now started

to develop ensemble classifiers designed to combine multiple individual classifiers while improving the classification performance of an IDS [2]. For instance, if a single classifier is trained on a set of subsets of an IDS dataset, it can produce different results. However, by implementing an ensemble model, it can achieve better performance. However, due to the complexity of the network traffic attributes and the number of attack types that can be considered, ML models are also prone to experiencing time and computational issues. One of the most effective ways to improve the performance of an IDS is by implementing feature selection [3,4]. This method can help the system identify highly relevant features and prevent useless ones from being detected.

This paper presents a novel approach to developing an IDS that can detect various types of attacks. It can be carried out through a Principal Component Analysis (PCA) method that eliminates redundancy and reduces the system's dimensionality. The proposed model shows how to improve the performance of an IDS by implementing an ensemble model. This method combines multiple decisions from multiple classifiers (Extra Trees (ET) [5,6], K Nearest Neighbors (KNN) [7,8], and Random Forest (RF)) into one model [9,10]. The model is based on an Average-of-Probabilities (AOP) vote combination rule.

The combination of PCA and the ensemble model can improve the accuracy and stability of an IDS by reducing its time and computational issues. It can also generate an unbiased model to perform better analysis. Furthermore, the system performance must be analyzed before deploying it in the real world. Thus, an adequate dataset must be available to evaluate the performance of IDS. Therefore, a dataset is chosen based on the training and testing of the model. However, only limited datasets are publicly accessible, which remains a challenge today, and a few among them even lack comprehensiveness and completeness. Network Security Laboratory–Knowledge Discovery in Databases (NSL–KDD) is the commonly used dataset for IDS [11]. IDS faces many challenges, including misjudgment, false detection, and the absence of real-time responses.

The paper is organized into five sections. Section 2 introduces the literature review about IDS solutions. The proposed methodology is demonstrated in Section 3. Section 4 shows, discusses, and analyzes the results of the experiments.

## 2. Literature Review

Several studies have been conducted to identify and detect attacks in smart cities. ML-based systems have proven to be effective in quickly detecting intrusion and working efficiently with a large amount of data, considering the destruction of the working principle and purpose of the system design. Feature selection must be considered to increase model performance and reduce data dimensionality by removing redundant or irrelevant features during the construction of an IDS.

In [12], the pigeon-inspired optimizer was introduced to identify features with a DT classifier to detect attacks by selecting the most critical features from the dataset. The researchers tried to compare feature selection techniques and evaluate their performance on three datasets: NSL–KDD, KDD cup'99, and UNSW–NB15. The study showed that the model's performance using NSL–KDD with 14 characteristics was 86.9% in accuracy. In [13], the RF classifier removed irrelevant traits from the dataset. Several ML models, such as KNN, Support-Vector Machine (SVM), Decision Tree (DT), and Logistic Regression (LR), were used to train and test the model. As a result, the model achieved an accuracy of 99.3% and 99.2% detection rate on selected significant features (=10) from an entire set (=41) in the NSL–KDD dataset. Hosseini [14] introduced an ML algorithm consisting of three main components: LR, genetic algorithm (GA), and artificial neural network (ANN). The first stage involved extracting the trait set from the data. The AI neural network (NN) was trained to detect intrusions in the second stage. The performance of the proposed model was 94.4% in terms of accuracy. Two datasets were used to analyze the proposed model. One was the NSL–KDD dataset, and the other was the KDD cup'99. Although the proposed model was only 88.90%, the training and testing time for the proposed method was shallow, as it was 74 s.

The differential evolution technique was introduced in [15] to reduce the number of traits in the data. This method mainly affects the accuracy of the intrusion detection system. The accuracy of the proposed model was 87.3% for the binary classification and 80.15% for multiple classifications. The proposed model was evaluated against two datasets: NSL–KDD and KDD cup'99. The proposed model performed lower in accuracy compared to existing studies in intrusion detection systems.

Iram et al. in [16] studied network data classification by implementing multiple ML technologies such as SVM, KNN, LR, Naive Bayes (NB), Multi-layer Perceptron (MLP), RF, DT, and ET. Study results were evaluated on four subsets of the NSL–KDD dataset, and an accuracy of over 99% was achieved using random forest classifiers, Extra Trees, and decision tree in all four feature subsets. To reduce features from the dataset and eliminate noise, De la Hoz et al. used PCA and Fisher Discriminant Ratio (FDR) [17]. The authors developed a probabilistic self-organizing map model to model feature space and identify normal patterns from anomalous ones. The accuracy, specificity, and sensitivity results were 90%, 93%, and 97%, respectively.

Moreover, the clustering method combines several basic models to reduce false positive rates and produce more accurate solutions. Several studies have been conducted using ensemble methods to utilize the advantages of more than individual classifiers in one model. The authors in [18] demonstrated how ensemble machine learning, NN, and kernel methods can detect abnormal behavior in an IoT IDS. In this study, ensemble methods outperformed NN and kernels in accuracy by 99.1%. The study was evaluated using the Kitsune and NSL–KDD datasets. In [19], cyberattacks using cluster methods for IoT-based smart cities were revealed. The authors evaluated their new model using the UNSW-NB15 and CICIDS2017 datasets, and in this study, the proposed method was found to be consistent with LR, SVM, DT, RF, ANN, and KNN with an accuracy of 99.91%. Zhou et al. in [20] proposed IDS considers the correlation between the attributes and uses a feature selection method, the CFS-BA, to reduce the data's dimensionality. The system was then implemented by combining C4.5 and RF algorithms through voting. The results showed that the system achieved 99.8% accuracy on the NSL–KDD dataset with 10 selected features. Even though the results showed adequate accuracy, the system was tested using only 10 validations, which is not a guarantee if the entire test dataset is used. Even though there have been a number of studies [21,22] in the field of IDS, a number of problems still need to be solved. An enhanced Long Short-Term Memory (LSTM) network was proposed by Elsayed et al. in [23] to distinguish between benign and malicious traffic, identify the attack category, and define the type of sub-attacks. ToN–IoT and InSDN datasets were used to assess the proposed system. The authors of [24] proposed a Transformer-based IoT NIDS method to determine the characteristics of attacks and their effects based on the types of data generated within the heterogeneous IoT environments. Using self-attention mechanisms, this method learns contextual embedding for input network features. It is capable of handling continuous and categorical features. The method uses network traffic data and telemetry information from IoT sensors to detect intrusions.

To tackle the same, a more diverse model was formed in the current work, where several classifiers that were not used in the prior research were combined to optimize the model performance. As such, a more robust model was obtained that could be effectively used for intrusion detection.

## 3. Methodology

To improve the detection capabilities of IDS, we propose an effective ML-based IDS using PCA. This method involves taking the data in a reduced form and keeping most of its original variance. The framework was built using voting, which is an ensemble of classifiers. The framework for developing an intelligent detection system (IDS) based on ML is shown in Figure 1.

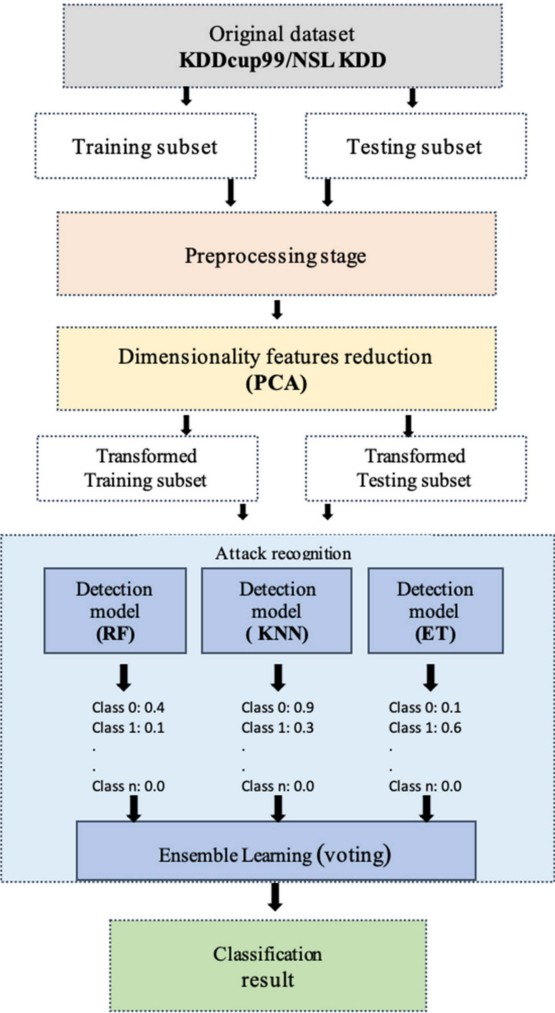

**Figure 1.** The framework of the PCA–Ensemble model.

In the dataset and preprocessing stage, the original dataset was processed to transform it into a suitable format for analysis. Then, the PCA method was applied in the dimensionality features reduction stage to scale down high-dimensional datasets by selecting the most appropriate features for each attack. Next, in the training classifier stage, three different classifiers were trained as base learners to improve the accuracy of the IDS using ET, KNN, and RF techniques. These classifiers were then used to create an ensemble classifier. After that, the attack recognition model was tested using a cross-validation approach and a voting technique to determine the probability of the base learners making the classification decision. Finally, benign traffic and various intrusive events could be classified and detected with high classification accuracy, according to the results of the ensemble classifier. Sections 3.1–3.3 provide detailed information about the stages of the proposed framework.

### 3.1. Dataset and Preprocessing

The NSL–KDD dataset retained the original dataset's characteristics, such as its advantageous and challenging structure. The new version of the dataset addressed some drawbacks inherited from the previous version, reduced the number of instances, and maintained the diversity of selected samples. The NSL–KDD dataset was compiled to maximize its difficulty of prediction. In order to classify the records according to their difficulty level, several benchmark classifiers were used [25]. The number of selected records for each difficulty level group was inversely proportional to records' percentage from the original dataset. The KDDTrain+, KDDTest+, and KDDTest-21 sets were used to classify the records in this study. The KDDTrain+ set comprises 125,973 instances composed of 67,343 instances

of normal traffic and 58,630 instances of attack traffic. The KDDTest+ set contains over 22,000 instances, while the KDDTest-21 set includes 11,850 instances.

The details of the datasets' instances are shown in Table 1. Each instance of the KDDTrain+ set has 42 attributes representing the different features of the connection. The values of these attributes are labeled as an attack or a normal.

**Table 1.** Statistics of the three sets of the NSL–KDD dataset.

| Class | NSL–KDD | | |
|---|---|---|---|
| | KDDTrain+ | KDDTest+ | KDDTest-21 |
| Normal | 67,343 | 9711 | 2152 |
| DoS | 45,927 | 7458 | 4342 |
| PRB | 11,656 | 2421 | 2402 |
| R2L | 995 | 2754 | 2754 |
| U2R | 52 | 200 | 200 |
| Attack | 58,630 | 12,833 | 9698 |
| Total | 125,973 | 22,544 | 11,850 |

NSL–KDD consists of four main attack types: denial-of-service (DOS), Remote to user (R2L), User to Root attack (U2R), and Probing attack (PRB).

The most critical step in data mining is preprocessing the raw data. This process involves extracting the necessary details from the data. Unfortunately, the data coming from heterogeneous platforms are often noisy, incomplete, and inconsistent. This is why it is essential to transform them into a format that can be used for knowledge discovery. The preprocessing step of this research involves analyzing and transforming the data. Due to the varying requirements of the platforms, the data may contain redundant and anomalous instances. This redundancy can affect the accuracy of classification. Therefore, all the records containing redundant values should be removed from the dataset at the start of the experiments to prevent data duplication.

Regarding symbolic values, for instance, in the NSL–KDD datasets, the feature "protocol type" includes values such as TCP, UDP, and ICMP protocols. Therefore, the conversion process is considered vital in order to improve the accuracy of IDS. In this paper, we replaced each symbolic feature with integer values. In addition, due to the varying scales of features, the classification performance can be affected by size. For instance, features with large numerical values can overwhelm the model's performance compared to features with small numerical values. Accordingly, we took normalization into account in our experiment.

### 3.2. Reduction of Dimensionality

A feature selection process aims to find a subset of attributes representing the data collected from an intrusion detection dataset [26]. These attributes ensure that the algorithm can interpret the data correctly. Unfortunately, many irrelevant and redundant attributes exist in modern intrusion detection datasets [27]. This study proposes a method that aims to reduce the dimensionality of the data and select the feature subset representative of the data collected from the dataset. It also aims to improve the accuracy of the classification process by implementing the PCA technique. The main idea of this method is to evaluate the relevance of the selected feature subset and the redundancy of the data in the given search space. It uses different PCAs with selected features. In PCA, original variables are transformed into k principal components, which capture data variance. Despite reducing the dimensionality of data and detecting patterns, PCA can have difficulty in interpreting the resulting principal components. Therefore, it is difficult to determine what features have been identified. A PCA is a technique that combines the results of multiple correlated variables into several uncorrelated ones. This method aims to transform these variables

into several principal components. The number of principal components derived from the various correlated variables is usually less than or equal to the original number of variables. Therefore, PCA aims to reduce the number of initial variables with significant dimensionality while retaining as much variance as possible. Let us consider a set of connection vectors composed of $v_1$, $v_2$, $v_3$, . . ., $v_M$. The following steps are used to calculate the PCAs of a dataset:

1. Assume the entire dataset has been obtained;
2. For each dimension, calculate the mean vector;
3. For the entire dataset, calculate the covariance matrix;
4. Identify the eigenvectors ($e_1$, $e_2$, $e_3$, . . . $e_d$) and eigenvalues ($v_1$, $v_2$, $v_3$, . . .. $v_d$);
5. Select the eigenvector with the highest eigenvalues and sort the eigenvalues in decreasing order;
6. By using this M form, a new sample space can be created;
7. A principal component is determined from the samples obtained.

### 3.3. Ensemble Classification

The method combines multiple base classifiers for ensemble learning to improve accuracy. This method can solve the same problem and produce much higher prediction results in stability and accuracy. The main reason ensemble classifiers are commonly used is their ability to improve the accuracy and performance of a given project. Another reason ensemble learning is commonly used is insufficient training data. This can lead to a weak or erroneous hypothesis. In this case, the individual classifier will spend significant time developing a reasonable hypothesis.

There are traditionally three categories of ensemble learning: voting, bagging, and boosting. The vote and bagging methods are similar in that they combine multiple algorithms to determine the final result, but they differ in how data are sampled. The boosting method differs slightly from the voting and bagging methods in ensemble learning. In the boosting method, models are trained sequentially, one after another.

Voting is the most popular method used in ensemble learning to improve classification performance by combining multiple classifiers' advantages into one model. It is widely used to build various models. Ensemble methods, such as intrusion detection, can often improve classification accuracy in security applications. Voting is more suitable for heterogeneous learners' ensembles (ET, KNN, RF) with lower computational complexity and less time overhead. ET has been widely used in anomaly detection among decision tree algorithms due to its high efficiency and superficial characteristics. The main advantage of KNN is that it can be applied to various programming problems, such as quadratic programming. This allows the current optimal solution to be continuously renewed. Random forest, on the other hand, is the most representative algorithm used in ensemble learning techniques. It is typically more reliable and can achieve better results than individual decision trees. As a result, ET, KNN, and random forest were chosen to build the ensemble for multi-class intrusion detection.

### 3.3.1. Extra Trees Classifier

The ET classifier aims to provide a prediction and classification framework for analyzing and predicting trees. When growing a tree in a random forest, the features that are considered for splitting are only random. This method can make the trees more random by considering the random thresholds found in each feature. An extremely random forest called an Extra Trees ensemble is a tree considered for classification and prediction. This also makes the training of Extra Trees faster since finding the optimal threshold for each feature at each node is very time-consuming. The prediction aims to determine the number of trees in a forest, and the selected features are random. Each tree in the forest represents a different class of prediction. This algorithm performs the random feature selection process on a case-by-case basis [28].

### 3.3.2. K-Nearest Neighbors Classifier

The KNN algorithm is a highly regarded machine learning and data-mining algorithm for classification. It is straightforward to implement and is suited to various tasks such as searching. The main reason it is considered one of the most influential classification methods is its ability to use various distance weighting measures. The KNN algorithm is mainly used for classification as it considers the various elements of a record set. For instance, the distance measures generally use Euclidian distance and the value of K number of neighbors. The type of KNN algorithm that is used for classification. It considers the training data of the various K nearest neighbors and predicts the class value of an unknown record with the help of its nearest neighbors [29,30]. The distance between the training data (point = $x$) and the testing data (point = $y$) is calculated using the Euclidean formula:

$$d(x,y) = \sqrt{\sum_{i=1}^{n} \int \left( (Xi : Yi) - (wi)^2 \right)} \tag{1}$$

where,

$x$ = training data
$y$ = data testing
$n$ = number of attributes
$f$ = similarity function between point x and point y
$wi$ = weight is given to attribute *i*.

### 3.3.3. Random Forest (RF) Classifier

Random Forest is a decision tree technique that constructs multiple decision trees that classify thousands of input variables based on their relevance. It can be viewed as an ensemble of classification trees that cast one vote for the class that appears most frequently in the data. Compared to other ML techniques, such as support vector machines or artificial neural networks, RF does not require many parameters to be specified. In RF, a collection of individual tree-structured classifiers can be defined as

$$\{h(x, \theta_k), k = 1, 2, \ldots i..\} \tag{2}$$

where $h$ is a RF classifier, $\{\theta_k\}$ represents identical random vectors that are independently distributed. Each tree gets a vote for every well-known class at input $x$. The nature and dimensionality of $\theta$ depend on how it is used during the construction of a tree.

The main goal of RF is to create a decision tree representing the forest. This is performed by training a subset of the training dataset, around two-thirds. Out of Bag (OOB) samples are elements employed for inner cross-validation to evaluate the RF's classification accuracy.

RF does not require many computational resources to perform its task, unlike other methods. It is also insensitive to outliers and parameters; therefore, it is unnecessary to prune the trees, which is a cumbersome task [31].

### 3.3.4. Voting Algorithm

A voting algorithm is a meta-model that performs the decision process by implementing several classifiers. It considers the factors influencing the decision and applies a combination rule to perform the final step. For instance, the algorithm combination rules are the product of probabilities, maximum, minimum, and average of probabilities.

Due to the number of classes in a classification, majority voting cannot be performed due to the complexity of the task. This paper introduces the average classifier of probabilities approach to perform the decision. The average of the predicted probabilities can determine the class label.

Suppose we have $l$ classifiers $C$, and c classes $\Omega = \{\omega_1, \ldots\ldots, \omega_c\}$. For instance, due to the classifiers considered in our experiment, $l$ can be set to 3, and the value of $c$ depends on the attack types. A classifier $C: R^n \to [0, 1]$ accepts an object $x \in R^n$ and outputs a vector:

$$[p_{c_i}(\omega_1|x), \ldots. p_c(\omega_c|x)] \tag{3}$$

where $p_c(\omega_j|x)$ is the probability set by the classifier to determine which object $x$ belongs to a class $\omega_j$. For each class $\omega_j$, let $m_i$ represent the mean of the probabilities assigned by the $l$ classifiers, which can be calculated as

$$m_i = \frac{1}{I}\sum_{i=1}^{I} p_{c_i}(\omega_j|x) \tag{4}$$

### 4. Results

The performance of the IDS was evaluated based on its ability to classify network traffic into a specific type. The paper presents the results of the testing process of the proposed algorithm, which was performed by the ensemble. We compared its performance by various metrics, such as accuracy, precision, detection rate (DR), F-measure, time (training and testing), and error rate. The first step in the PCA process was to identify the PCAs. The proposed PCA method can reduce the dimensionality of the dataset significantly. It also eliminates irrelevant features. An ensemble classifier was also employed to increase the performance of IDS. This method combines three classifiers in a voting algorithm: RF, KNN, and ET.

As a result, four separate classifiers were built using the training and testing datasets for classification. Table 2 shows the best classification performance with and without the dimensionality reduction method regarding the primary metrics used.

**Table 2.** (a) The performance outcomes according to original features. (b) The performance outcomes according to the chosen features using PCA.

| (a) | | | | | | | | |
|---|---|---|---|---|---|---|---|---|
| **Classifier** | **PCAs** | **Accuracy** | **Precision** | **DR** | **F-Measure** | **Training (s)** | **Testing (s)** | **Error Rate** |
| Ensemble | * | 0.996 | 0.999 | 0.999 | 0.999 | 36.22 | 3.9 | 0.004 |
| RF | * | 0.997 | 0.999 | 0.999 | 0.999 | 5.493 | 0.613 | 0.003 |
| KNN | * | 0.991 | 0.995 | 0.995 | 0.995 | 2157.8 | 231.024 | 0.003 |
| ET | * | 0.997 | 0.998 | 0.998 | 0.998 | 0.113 | 0.014 | 0.003 |
| PSOM [17] | * | 0.88 | 0.90 | 0.92 | - | - | - | - |
| **(b)** | | | | | | | | |
| **Classifier** | **PCAs** | **Accuracy** | **Precision** | **DR** | **F-Measure** | **Training (s)** | **Testing (s)** | **Error Rate** |
| Ensemble | 30 | 0.998 | 0.998 | 0.998 | 0.998 | 19.256 | 2.238 | 0.002 |
| RF | 24 | 0.978 | 0.987 | 0.977 | 0.977 | 4.490 | 0.518 | 0.003 |
| KNN | 24 | 0.926 | 0.966 | 0.986 | 0.976 | 14.280 | 1.600 | 0.004 |
| ET | 15 | 0.954 | 0.944 | 0.984 | 0.984 | 0.061 | 0.007 | 0.006 |
| PSOM+PCA [17] | 15 | 0.90 | 0.80 | 0.97 | - | - | - | - |

The number of PCAs in (b) part of Table 2 represents the principal components for each classifier.

We repeatedly ran the experiment to evaluate the performance of the classifiers. Gradually, the number of PCAs/features was increased for each classifier. In each iteration, we

increased the number of PCAs until adding a new one did not improve the model's performance.

As shown in Figure 2, the performance of the Ensemble models was not improved after 30 PCAs and 24, 24, and 15 PCAs for RF, KNN, and ET, respectively. The performance of the ensemble method obtained the maximum accuracy rate of 99.89% with 30 PCAs and exceeded all other individual classifiers. In contrast, the best accuracy of the RF, KNN, and ET classifiers were 97.85%, 92.65%, and 95.45% with 24, 24, and 15 PCAs, respectively.

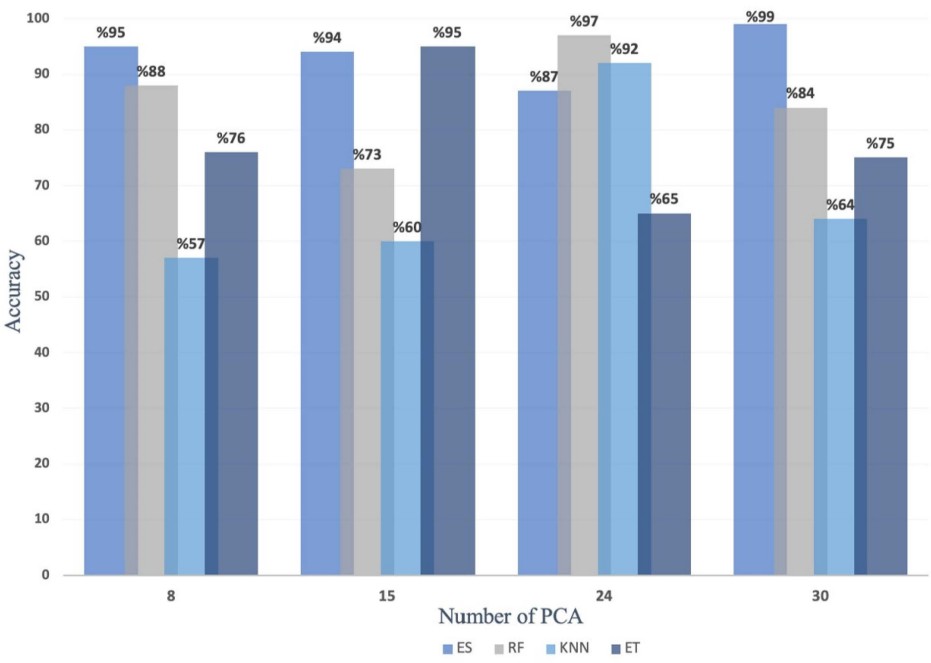

**Figure 2.** Performance of the ensemble models.

Moreover, the proposed model has the highest scores in DR, precision, and F-measure and the lowest error rate compared to other combined models, as shown in Table 2.

As compared to other studies, the results obtained using the ensemble method are superior when compared with other studies. With all the features included in the proposed method, the accuracy was 99%, while [17] achieved an accuracy of 88% as shown in (a) part of Table 2. In (b) part of Table 2, the proposed method achieved 99% accuracy by using the PCA dimension reduction method with 30 features. In comparison, the [17] method achieved 90% accuracy with 15 features when it was applied to the PCA dimension reduction method. A proposed model based on 15 features achieved 94% accuracy compared to a model based on the same number of features in [17], which achieved 90% accuracy. The dimensional reduction algorithm significantly reduces the computational cost when applied to the ensemble model. (a) part of Table 3 compares the training and testing times to the features used. The ensemble model with PCA reduced the training and testing times compared to the same model using all features. The ensemble model significantly mitigated the training and testing times, in seconds, from 36.22 and 3.58 to 19.25 and 2.238, respectively.

As a result of the specified PCA numbers, (b) part of Table 3 illustrates the execution time for each phase of the learning process and the testing process based on the specified PCA numbers. With the reduction in the number of PCAs, there is a reduction in the time needed to conduct the process. It is also noticed that the performance of most classifications is qualified. At the same time, several attacks cannot be classified very well, as seen in Figure 3. The numbers of 'U2R' and 'Heartbleed' are less than others, significantly affecting attack classification. In particular, there are only 52 'U2R' instances in the KDDTrain+ collection, making it difficult for the IDS to be classified correctly.

**Table 3.** (a) The computational time of classifiers; (b) The computational time of ensemble classifier with the number of PCA.

| (a) | | | | |
|---|---|---|---|---|
| | **Without PCAs** | | **With PCAs** | |
| | **Training (s)** | **Testing (s)** | **Training (s)** | **Testing (s)** |
| Ensemble | 36.22 | 3.9 | 19.256 | 2.238 |
| RF | 5.4932 | 0.61308 | 4.4909 | 0.518 |
| KNN | 2157.8 | 231.024 | 14.2809 | 1.600 |
| ET | 0.1132 | 0.014409 | 0.0617 | 0.0073 |

| (b) | | |
|---|---|---|
| **Number of PCA (Ensemble)** | **Training (s)** | **Testing (s)** |
| 10 | 13.403 | 1.604 |
| 15 | 14.79 | 1.742 |
| 20 | 16.081 | 1.894 |
| 25 | 17.48 | 2.058 |
| 30 | 19.256 | 2.238 |
| 35 | 20.793 | 2.537 |

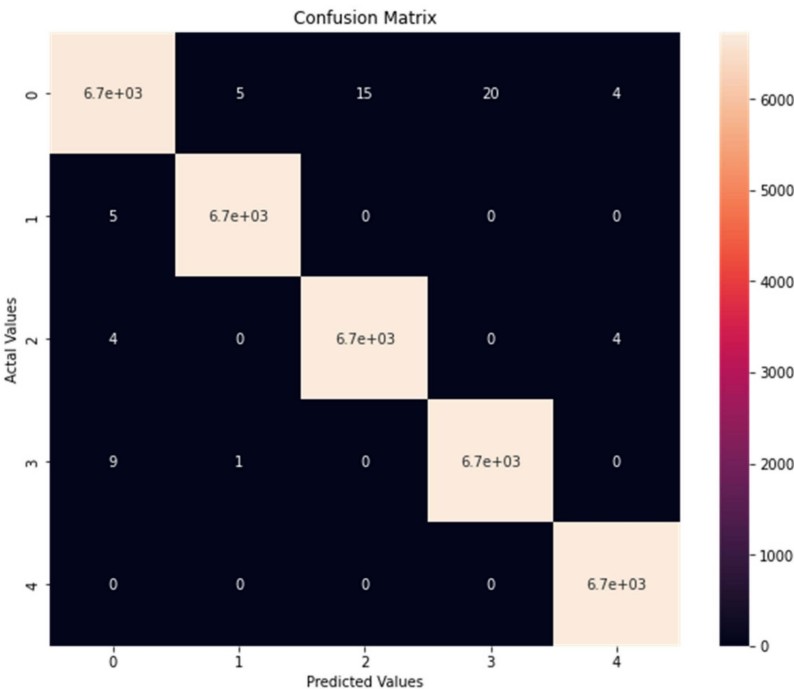

**Figure 3.** Confusion matrix of proposed IDS with 30 PCAs.

The proposed ensemble model chooses pertinent features for all classes and does not focus on a specific class. It does not guarantee the effectiveness of all attacks, particularly those with few instances in the datasets. However, the developed model can detect intrusions as the classification findings are relatively consistent across all datasets. The performance of the proposed IDS was evaluated by comparing it with the proposed PCA method with and without feature selection, as shown in Figure 4. The results of the study show that the proposed IDS with PCA outperforms the others when it comes to distinguishing benign instances from attacks. The average values of various metrics, such as accuracy, precision, and DR, have increased significantly.

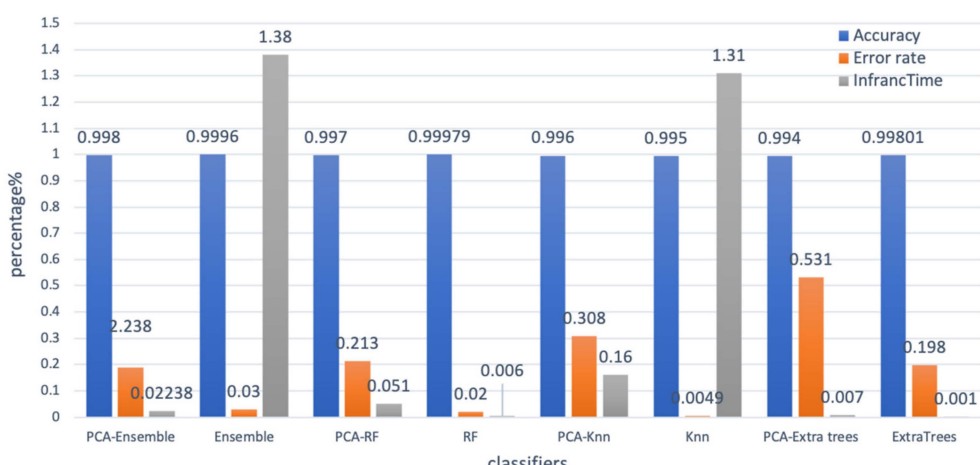

**Figure 4.** Comparison of performance of classifiers.

## 5. Conclusions

This paper proposed an approach to developing a multi-attack intrusion detection system using machine learning ensembles to overcome individual classifier weaknesses. KDD'99 was used to evaluate the system's performance in multiple network environments. The proposed model reduces system dimensionality and eliminates redundancy by employing Principal Component Analysis. Results indicate that the stacked ensemble-based model maximizes performance by combining various classifiers best suited to the given task. The accuracy of ensemble classifiers increases when dealing with highly imbalanced datasets in intrusion detection systems. The model generally exceeds expectations, but some areas can be improved. These areas include detecting attacks such as U2R and R2L in KDD'99. A network intrusion detection system with a high-class imbalance was investigated using an ensemble method. As well as improving class imbalance problems with synthetic oversampling, cost-sensitive learning models can sometimes enhance class detection. As quantum machine learning appears to have considerable promise, it may be possible in the future to improve the performance of the intrusion detection system by using a quantum approach in order to detect various types of attacks. Future research should explore whether this could be achieved.

**Author Contributions:** Conceptualization, M.A. and K.A.-S.; methodology, M.A.; software, M.A.; validation, M.A. and K.A.-S.; formal analysis, M.A.; investigation, K.A.-S.; resources, K.A.-S.; data curation, M.A.; writing—original draft preparation, M.A.; writing—review and editing, K.A.-S.; visualization, K.A.-S.; supervision, K.A.-S.; project administration, K.A.-S.; funding acquisition, K.A.-S. All authors have read and agreed to the published version of the manuscript.

**Funding:** The authors gratefully acknowledge Qassim University, represented by the Deanship of Scientific Research, on the financial support for this research under the number (COC-2022-1-1-J-25084) during the academic year 1444 AH/2022 AD.

**Acknowledgments:** The researchers would like to thank the Deanship of Scientific Research, Qassim University for funding the publication of this project.

**Conflicts of Interest:** The authors declare no conflict of interest.

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
