# Peer review of "Developing an Integrated Framework for Securing Internet of Things Traffic in Smart Cities Using Machine Learning Techniques"

_applsci, doi:10.3390/app13169476_

Round 1

Reviewer 1 Report

To improve the detection capabilities of IDS, the authors propose an effective ML-based IDS 148 using PCA

1. In Figure 1. The framework of the PCA-Ensemble model., you have not mentioned the role of the ensemble classifier. Whether you follow "Boosting or Bagging" foe classification. The framework may be redrawn to give much more clarity for the flow of the process

2. Another clarification needed is the dataset. KDD dataset is a fine tuned data set and there is no noise exists. Then how you can detect any security breach or any damage to the data.

3. Section 4 Results: Your method combines three classifiers in a voting algorithm RF, KNN, and ET. In table 2 and 3 you tabulated the values of the performance metrics of this three and Ensemble. Why this ensemble is added extra. You have combined RF, KNN, ET as ensemble. Then what is that "ensemble"

means. 

4. Conclusion

It starts with "This paper aims to present". However, the conclusion is the result and need not have any aim in this section. Need to rewrite the  conclusion part.

5. You may summarize the advantages and limitations of the papers in the Literature Review section. The following papers may also be references to improve the strength of the literature

doi: 10.1109/ICCONS.2017.8250578.

doi: 10.1109/IC3I56241.2022.10073271

doi.org/10.3837/tiis.2020.09.005

Reviewer 2 Report

In the era of IoT, the massive amount of traffic exchanged among devices may lead to potential security threats and risks. Hence, detecting suspicious traffic for secure data communication is crucial. The authors present a novel approach by combining PCA and an ensemble model to develop an IDS capable of detecting various types of attacks. The overall performance of the intrusion detection system is significantly improved. I believe this paper is suitable for publication in applied sciences. However, I have the following comments for the authors to address before this manuscript can be accepted:

1.       The authors should ensure that Table 2. (a) and Table 2. (b) have consistent data formats and precision for easy comparison.

2.       It is suggested that the authors compare the performance outcomes presented in Table 2. (a) and Table 2. (b) with other related works. Adding related discussions will be beneficial.

3.       In the sentence, "The ensemble model 352 significantly mitigates the training and testing times from 36.22 and 3.58 to 19.25 and 353 2.238, respectively," on Page 9, the time unit of seconds should be added.

4.       Clarify the meaning of the number of PCAs in Table 2. (b). Does "the number of PCAs" represent the number of features selected by PCA? If yes, the authors are encouraged to provide details of the selections and demonstrate the specific features selected.

5.       Improve the captions of all figures and tables to make them less vague. Add relevant explanations to help readers easily understand the figures and tables. For instance, explain the meanings of values appearing in the black squares of Figure 3 and why "30 PCAs" was chosen to demonstrate the confusion matrix.

6.       Consider if the number of PCAs affects the computational time of this method and include this information in the discussion.

7.       Mention that quantum machine learning is a promising direction and suggest exploring if a quantum model can enhance the performance of the intrusion detection system to detect various attack types. Refer to recent interesting works in this area. The authors can make some discussion.

[r1] Toward quantum advantage in financial market risk using quantum gradient algorithms, Quantum 6, 770 (2022).

[r2] Quantum Neural Network for Quantum Neural Computing, Research 6, 0134 (2023).

[r3] Quantum principal component analysis, Nature Physics 10, 631-633 (2014).

[r4] Experimental quantum advantage with quantum coupon collector, Research 2022, 9798679 (2022).

Moderate editing of English language required

Reviewer 3 Report

The manuscript has organized the research content in a relatively complete manner, but it needs to highlight its innovativeness.

1. The literature research is not sufficient. There are other studies on the intrusion detection method using Average-of-Probabilities for ensemble learning. Why is there no citation analysis and comparison?

2. There is an inappropriate floating box in Figure 4, and the image needs to be regenerated.

3. The manuscript needs to explain the difference from other literature in the introduction or Chapter 2, as ensemble methods and PCA are common research methods.

4. The composition of the table needs to be introduced in detail, such as the meaning of “Testing” and “Buding” in Table 2.

5. Ensure that the capitalization in Figure 2 is correct and consistent with the manuscript.

6. The format of the references needs to be consistent, for example, the 8th, 9th, and 10th formats are inconsistent.

Round 2

Reviewer 1 Report

All the comments are incorporated in the paper. 

Author Response

No comments!

Thank you for your efforts.

Reviewer 2 Report

The author has solved most of my issues.  But some comments remain unresolved. 

1.  It is suggestedthat the authors compare the performance outcomes presented in Table 2. (a) and Table 2. (b) with other related works. Adding related discussions will be beneficial.  "Ascompared to other studies, the results obtained using the ensemble method aresuperior when compared with other studies. With all the features included in the proposed method, the accuracy was 99%, while [17] achieved an accuracy of 88% as shown in Table 2 (a). In Table 2 (b), the proposed method achieved 99% accuracy by using the PCA dimension reduction method with 30 features. In comparison, the [17] method achieved 90% accuracy with eight features when it was applied to the PCA dimension reduction method"

However,the compared work[17] has less  selected features. We can deduce that the advantages of authors’s method merely orginite form using more features in PCA.

2. Clarify the meaning of the number of PCAs in Table 2. (b). Does "the number of PCAs" represent the number of features selected by PCA? If yes, the authors are encouraged to provide details of the selections and demonstrate the specific features selected. "The proposed model uses different PCAs with several selected features. In PCA, original variables are transformed into k variables called principal components. These variables capture data variance. The principal component analysis (PCA) reduces data dimensionality and detection patterns, but the resulting principal components are not always easy to interpret. Thus, it is difficult to determine what features have been identified."

Suggest adding this related  discussion into the manuscript.

3. Mention that quantum machine learning is a promising direction and suggest exploring if a quantum model can enhance the performance of the intrusion detection system to detect various attack types.  To let the reader understand the relevant background and development dirction, the authors can make some discussion.

  •  

 Minor editing of English language required

Author Response

1. However, the compared work[17] has less  selected features. We can deduce that the advantages of authors’ method merely orginite form using more features in PCA.

A proposed model based on 15 features achieved 94% accuracy compared to a model based on the same number of features in [17], which achieved 90% accuracy.

2. Suggest adding this related  discussion into the manuscript.

Done

3. Mention that quantum machine learning is a promising direction and suggest exploring if a quantum model can enhance the performance of the intrusion detection system to detect various attack types. To let the reader understand the relevant background and development direction, the authors can make some discussion.

As quantum machine learning appears to have considerable promise, it may be possible in the future to improve the performance of the intrusion detection system by using a quantum approach in order to detect various types of attacks. Future research should explore whether this could be achieved.

Reviewer 3 Report

1. The format of the references in the re-uploaded manuscript is still inconsistent. For example, the journal names in references 8 and 14 are abbreviated, while the first letters of the journal names in references 10 and 31 are not capitalized, while others are capitalized.

2. The manuscript still lacks sufficient introduction and comparison of references. Please search for the latest relevant literature using “intrusion detection system” and “Average-of-Probabilities” as keywords, and introduce or compare them to highlight the novelty of the manuscript’s work.

3. The meaning of DR in Table 2 is still not clearly introduced.

Author Response

1. The format of the references in the re-uploaded manuscript is still inconsistent. For example, the journal names in references 8 and 14 are abbreviated, while the first letters of the journal names in references 10 and 31 are not capitalized, while others are capitalized.

All references have been checked and corrected.

2. The manuscript still lacks sufficient introduction and comparison of references. Please search for the latest relevant literature using “intrusion detection system” and “Average-of-Probabilities” as keywords, and introduce or compare them to highlight the novelty of the manuscript’s work.

IDS average-of-probability is another study goal that requires a research paper in this field.

3. The meaning of DR in Table 2 is still not clearly introduced.

Detection Rate (DR) has been added in the text.